# A molecular barcode and web-based data analysis tool to identify imported *Plasmodium vivax* malaria

Hidayat Trimarsanto[1,2], Roberto Amato[3], Richard D. Pearson[3], Edwin Sutanto[2,4], Rintis Noviyanti[2], Leily Trianty[2], Jutta Marfurt[1], Zuleima Pava[1], Diego F. Echeverry[5,6,7], Tatiana M. Lopera-Mesa[8], Lidia M. Montenegro[8], Alberto Tobón-Castaño[8], Matthew J. Grigg[1,9], Bridget Barber[1,9], Timothy William[9,10], Nicholas M. Anstey[1], Sisay Getachew[11,12], Beyene Petros[11], Abraham Aseffa[12], Ashenafi Assefa[13], Awab G. Rahim[14,15], Nguyen H. Chau[16], Tran T. Hien[16], Mohammad S. Alam[17], Wasif A. Khan[17], Benedikt Ley[1], Kamala Thriemer[1], Sonam Wangchuck[18], Yaghoob Hamedi[19], Ishag Adam[20], Yaobao Liu[21,22], Qi Gao[21], Kanlaya Sriprawat[23], Marcelo U. Ferreira[24,25], Moses Laman[26], Alyssa Barry[27,28,29], Ivo Mueller[28,30], Marcus V. G. Lacerda[31,32], Alejandro Llanos-Cuentas[33], Srivicha Krudsood[34], Chanthap Lon[35], Rezika Mohammed[36], Daniel Yilma[37], Dhelio B. Pereira[38], Fe E. J. Espino[39], Cindy S. Chu[14,23], Iván D. Vélez[8], Chayadol Namaik-larp[40], Maria F. Villegas[41], Justin A. Green[42], Gavin Koh[42], Julian C. Rayner[3,43], Eleanor Drury[3], Sónia Gonçalves[3], Victoria Simpson[3], Olivo Miotto[3,14], Alistair Miles[3], Nicholas J. White[14,44], Francois Nosten[23,44], Dominic P. Kwiatkowski[3], Ric N. Price[1,14,44] & Sarah Auburn[1,14,44✉]

Traditionally, patient travel history has been used to distinguish imported from autochthonous malaria cases, but the dormant liver stages of *Plasmodium vivax* confound this approach. Molecular tools offer an alternative method to identify, and map imported cases. Using machine learning approaches incorporating hierarchical fixation index and decision tree analyses applied to 799 *P. vivax* genomes from 21 countries, we identified 33-SNP, 50-SNP and 55-SNP barcodes (GEO33, GEO50 and GEO55), with high capacity to predict the infection's country of origin. The Matthews correlation coefficient (MCC) for an existing, commonly applied 38-SNP barcode (BR38) exceeded 0.80 in 62% countries. The GEO panels outperformed BR38, with median MCCs > 0.80 in 90% countries at GEO33, and 95% at GEO50 and GEO55. An online, open-access, likelihood-based classifier framework was established to support data analysis (vivaxGEN-geo). The SNP selection and classifier methods can be readily amended for other use cases to support malaria control programs.

---

A full list of author affiliations appears at the end of the paper.

The last three World Malaria Reports have revealed a disturbing rise in malaria cases, and, outside Sub-Saharan Africa, an increasing proportion of malaria due to *Plasmodium vivax*, undermining the concerted efforts to reduce transmission over the past decade[1]. These trends highlight the urgent need for new surveillance tools, and the need for greater attention to non-falciparum *Plasmodium* species. One particular challenge for malaria control are highly mobile human populations, leading to the import of *Plasmodium* isolates from one country to another (imported cases) which can hinder local control efforts and enhance the risks of outbreaks and the spread of antimalarial drug resistance. To counteract this challenge there is a critical need to develop tools that can help to determine where patients acquired their infection.

Distinguishing between local and imported infection is particularly challenging for *P. vivax*, in view of the parasite's ability to form dormant liver stages (hypnozoites) that can reactivate weeks to months after the initial infection, as well as its ability to cause highly persistent, splenic and low-density circulating blood-stage infections that can evade routine diagnosis[2–4]. The re-emergence of *P. vivax* in multiple regions where it was once almost eliminated highlights the importance of diligent surveillance[5,6]. In low endemic settings, the relative proportion of imported cases generally rises as incidence falls, emphasizing the importance for surveillance tools that can identify imported *P. vivax* cases in these regions in particular[5]. Traditionally, imported cases have been identified and mapped using information on patient travel history, but the persistent splenic and blood stage infections and late relapses limit the accuracy of this approach for *P. vivax*. Molecular tools to identify and map imported *P. vivax* cases offer an attractive complement to traditional epidemiological tools.

Amplicon-based sequencing has become a favored approach for targeted genotyping of malaria parasites[7,8]. Using highly parallel sequencing platforms, such as the latest generation of Illumina sequencers, amplicon-based sequencing can be applied at moderate to high-throughput, with high accuracy and sensitivity. These platforms are flexible, allowing iterative enhancement of the Single Nucleotide Polymorphism (SNP) barcodes, which can provide an affordable genotyping approach, amenable to population-based molecular surveillance.

Previous studies have used mitochondrial and apicoplast markers to distinguish imported from local *P. vivax* isolates, but the resolution of these organellar genomes is constrained[9–11]. In 2015, a panel of 42 SNPs, commonly referred to as the Broad barcode, was identified to facilitate parasite finger-printing and geographic assignment[12]. The 42-SNP Broad barcode was derived from genomic data available from 13 isolates from 7 countries and has been applied to several studies using targeted genotyping assays[12–14]. A more recent study identified another *P. vivax* SNP barcode using data from 433 isolates from 17 countries[15]. This barcode also aimed to facilitate both finger-printing and geographic assignment, but no experimental assays for this barcode are available and it remains an in-silico tool only[15]. Furthermore, all geographic barcoding studies of malaria to date have relied on visual methods such as Principal Components Analysis to evaluate the country of origin. Whilst this approach has some utility, it is moderately subjective and does not cater to the needs of translational end users such as National Malaria Control Programs (NMCPs), who may not have the genetic epidemiology or bioinformatic skills required to generate and interpret these plots.

The primary objectives of our study were to establish a framework to identify *P. vivax* molecular markers for identifying and characterizing imported *P. vivax* cases by classifying country of origin and to develop an online, open-access informatics platform for end-users to analyze data generated using the markers. Our goal is for these new molecular and informatics tools to support the generation of evidence that can be used by both researchers and NMCPs to inform strategic decisions on where and how to deploy malaria control interventions. Our molecular tools are tailored primarily to surveillance frameworks using sequencing platforms such as Illumina or MinION (Oxford Nanopore Technologies), which enable genotyping of dozens of markers in parallel. Our informatics tools are designed to enable users with little or no genetics or bioinformatics skills to independently analyze and interpret barcode genotyping data generated in their country or at regional reference laboratories. The informatics tools are therefore designed to accommodate real-world malaria samples including polyclonal infections and samples with incomplete data resulting from genotyping failures.

## Results

**Data preparation for the training dataset**. The primary dataset (Dataset 1) that was derived using the missing data simulations to minimize genotype failures (Supplementary Fig. 1) comprised 229,317 high-quality informative SNPs and 826 high-quality samples. The median percentage of heterozygous calls in each sample ranged from 0.02% to 0.08%. Details on the geographic locations of the samples in Dataset 1 are presented in Supplementary Table 1. Using country-level assignments derived from the genome-wide data classification with the likelihood classifier, 27 isolates presented country classifications differing from the country of presentation, so are potentially imported cases (Supplementary Table 1). After exclusion of these cases, as well as countries represented by only a single sample, there were a total of 799 isolates from 21 countries, constituting Dataset 2 (Supplementary Table 1). Neighbor-joining analysis revealed distinct geographic clustering of most countries (Supplementary Fig. 2). Exceptions included the isolates from Afghanistan, Iran, India and Sri Lanka, which appeared to form a single cluster; further analysis of this geographic region with larger sample sets is clearly required to resolve inter-country differences. Although several isolates in border regions including Vietnam relative to Cambodia, and Thailand relative to Myanmar, overlapped between countries, most isolates in these countries could be differentiated by national boundaries.

**Candidate SNP panel selection**. The SNP panel selection process is summarized in Fig. 1. When the HFST selector was applied with an FST threshold of 0.90 (HFST-0.90), a set of 33 new candidate SNPs (herein referred to as GEO33) for geographic assignment were identified (Supplementary Table 2). On increasing the FST threshold to 0.95 (HFST-0.95), the HFST model identified 50 SNPs (herein referred to as GEO50) (Supplementary Table 3). Using the DT selector alone, 55 SNPs (herein referred to as GEO55) (Supplementary Table 4) were identified. As illustrated in Supplementary Fig. 3 and Supplementary Table 5, there is no marker overlap between the 38-SNP Broad panel (herein referred to as BR38) and the three new SNP panels, but varying levels of SNP overlap are present between the three new panels. Three SNPs are present in all three panels; a variant at PvP01_09_v1: 1884013 in the *IMC1b* gene (*PVP01_0942600*) that causes an E141D amino acid change, a variant at PvP01_10_v1:480601 in the *MDR1* gene (*PVP01_1010900*) that causes an L845F amino acid change, and a variant at PvP01_14_v1:1229487 in PVP01_1428700 that causes and S1136I amino acid change. A further 6 SNPs overlapped between the GEO33 and GEO50 panels, and 13 SNPs overlapped between the GEO50 and GEO55 panels. Amongst the SNPs overlapping between two panels, the most notable is a variant at PvP01_14_v1:1270401 in the *PPPK-DHPS* gene (*PVP01_1429500*) that causes an A553G amino acid change that has been associated with sulfadoxine resistance[16].

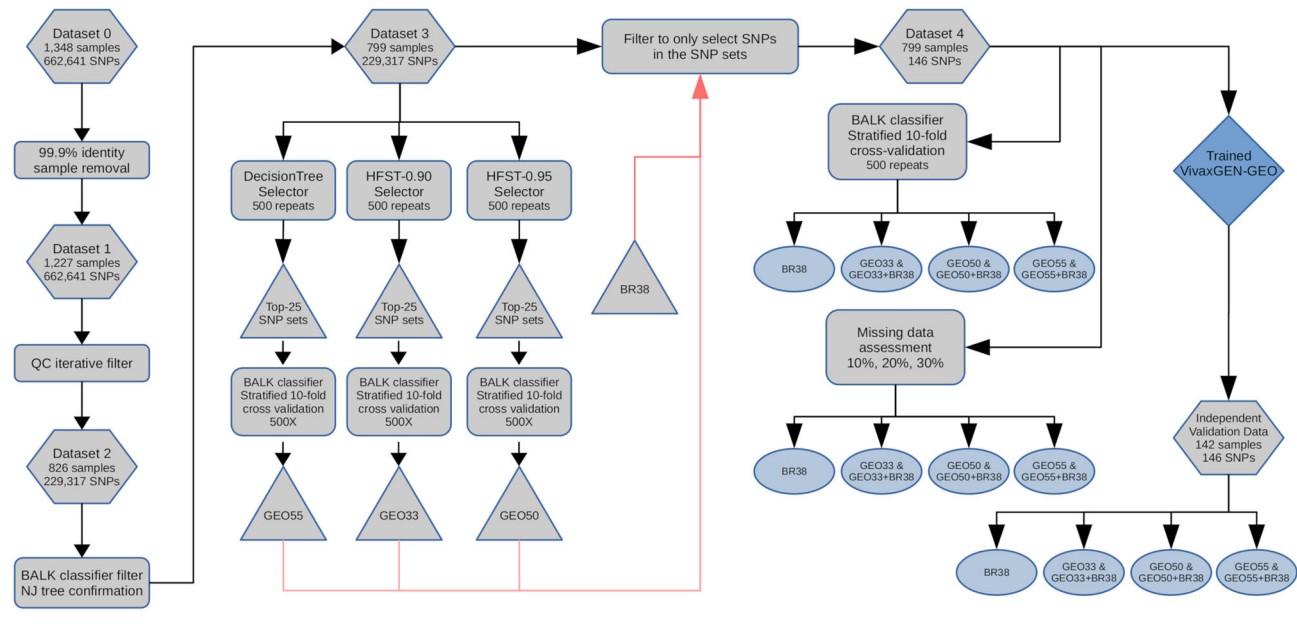

**Fig. 1 Overview of the sample processing and marker selection approaches.** Hexagons reflect datasets, rectangles reflect processes, triangles reflect SNP sets, ovals reflect results, and the diamond reflects the web-based classifier application. The BR38 Broad barcode reflects 38 assayable SNPs of the 42 Broad SNPs. The GEO33 set reflects the high-performance SNPs derived from the HFST approach with FST threshold of 0.9. The GEO50 set reflects the high-performance SNPs from the HFST approach with threshold FST of 0.95. The GEO55 set reflects the SNPs selected by the Decision Tree approach.

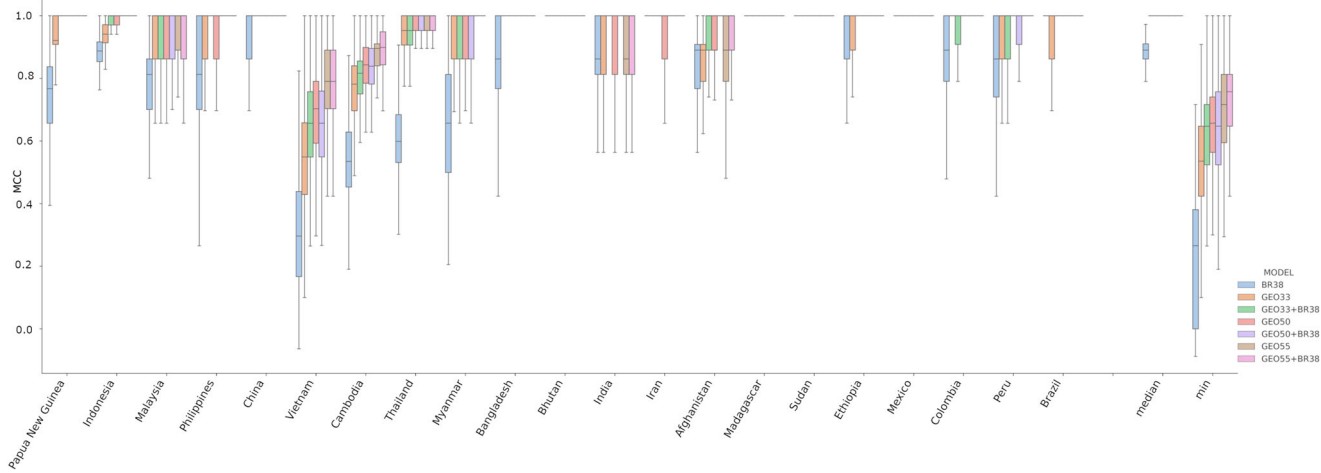

**Fig. 2 Comparison of country prediction performance between the SNP panels.** The boxplots present the MCC scores from 500 repeats with stratified 10-fold cross validation for each SNP set. Country labels are provided on the y-axis; median and min reflect the respective summary statistics for the pooled MCC scores across all countries. Each bar presents the median, interquartile range and min and max MCC for the given country and model. The BR38 panel generally exhibited the lowest MCC scores (i.e., lowest prediction accuracy). Amongst the newly selected panels, GEO55 generally gave the highest MCC scores followed by GEO50, and then GEO33. The addition of the BR38 panel to the GEO panels generally only provided modest if any increase in the median MCC. The analyses were based on $n = 799$ biologically independent samples.

**Comparative assessment of candidate SNP panels with no missing data.** The classification performance of the BR38, GEO33, GEO50, GEO55, and combinations of BR38 with the three new GEO panels (i.e., GEO33 + BR38, GEO50 + BR38 and GEO55 + BR38) was analyzed by 10-fold cross-validation using the BALK classifier on the samples in Dataset 3. The results of the evaluations in Dataset 3 are illustrated in Fig. 2 (source data provided in Supplementary Data 1), and the median MCCs reflecting the consensus results of the cross-validation are summarized in Table 1. The BR38 barcode exhibited the lowest pooled (country wide) median MCC (median MCC = 0.84),

followed by GEO33 (median MCC = 0.94) and GEO50 and GEO55 (both median MCC = 1.00). The pooled median MCCs for the combined GEO and BR38 panels all exceeded 1.00 but provided only minor improvements for GEO50 and GEO55. The percentage of countries exhibiting median MCCs greater than 0.8 was 62% (13/21) at BR38, 90% (19/21) at GEO33 and GEO33 + BR38, and 95% (20/21) at GEO50, GEO55, GEO50 + BR38 and GEO55 + BR38. The countries with the lowest prediction performance were Vietnam and Cambodia. Vietnam exhibited median MCCs < 0.8 with all SNP panels. Cambodia exhibited median MCCs < 0.8 at BR38, GEO33 and GEO33 +

**Table 1 Summary of MCC scores from the results of 500 repeats of the stratified 10-fold cross-validation of the SNP panels.**

| Country | BR38 | GEO33 | GEO33 + BR38 | GEO50 | GEO50 + BR38 | GEO55 | GEO55 + BR38 |
|---|---|---|---|---|---|---|---|
| Afghanistan | 0.835 | 0.850 | 0.946 | 0.908 | 0.946 | 0.889 | 0.940 |
| Bangladesh | 0.784 | 1.000 | 1.000 | 1.000 | 1.000 | 1.000 | 1.000 |
| Bhutan | 1.000 | 1.000 | 1.000 | 1.000 | 1.000 | 1.000 | 1.000 |
| Brazil | 0.892 | 0.892 | 1.000 | 1.000 | 1.000 | 1.000 | 1.000 |
| Cambodia | 0.520 | 0.756 | 0.794 | 0.823 | 0.816 | 0.870 | 0.874 |
| China | 0.864 | 1.000 | 1.000 | 1.000 | 1.000 | 1.000 | 1.000 |
| Colombia | 0.870 | 1.000 | 1.000 | 1.000 | 1.000 | 1.000 | 1.000 |
| Ethiopia | 0.923 | 1.000 | 1.000 | 1.000 | 1.000 | 1.000 | 1.000 |
| India | 0.812 | 0.864 | 0.892 | 0.892 | 1.000 | 0.864 | 0.864 |
| Indonesia | 0.913 | 0.939 | 0.985 | 0.985 | 1.000 | 1.000 | 1.000 |
| Iran | 1.000 | 1.000 | 1.000 | 0.892 | 1.000 | 1.000 | 1.000 |
| Madagascar | 1.000 | 1.000 | 1.000 | 1.000 | 1.000 | 1.000 | 1.000 |
| Malaysia | 0.794 | 0.933 | 0.933 | 0.933 | 0.933 | 0.933 | 0.933 |
| Mexico | 1.000 | 1.000 | 1.000 | 1.000 | 1.000 | 1.000 | 1.000 |
| Myanmar | 0.620 | 0.910 | 0.892 | 0.910 | 0.892 | 0.892 | 1.000 |
| Papua New Guinea | 0.344 | 0.812 | 0.892 | 1.000 | 1.000 | 1.000 | 1.000 |
| Peru | 0.817 | 0.940 | 0.940 | 1.000 | 1.000 | 1.000 | 1.000 |
| Philippines | 0.663 | 0.892 | 1.000 | 0.864 | 1.000 | 1.000 | 1.000 |
| Sudan | 1.000 | 1.000 | 1.000 | 1.000 | 1.000 | 1.000 | 1.000 |
| Thailand | 0.588 | 0.927 | 0.952 | 0.975 | 0.955 | 0.976 | 0.976 |
| Vietnam | 0.277 | 0.520 | 0.617 | 0.676 | 0.648 | 0.769 | 0.755 |
| Pooled median | 0.835 | 0.939 | 1.000 | 1.000 | 1.000 | 1.000 | 1.000 |
| Pooled min | 0.277 | 0.520 | 0.617 | 0.676 | 0.648 | 0.769 | 0.755 |
| Pooled Q1 | 0.663 | 0.892 | 0.933 | 0.908 | 0.955 | 0.933 | 0.976 |

The analyses were based on $n = 799$ biologically independent samples.

BR38. Six countries (Philippines, Myanmar, Malaysia, Thailand, Papua New Guinea and Bangladesh) exhibited median MCCs < 0.8 with BR38 but exceeded 0.8 in all the GEO combinations.

**Comparative assessment of candidate SNP panels with missing data simulations**. To compare the performance of the BR38 barcode, GEO33, GEO50, GEO55 and combinations of BR38 with the three new GEO panels (i.e., GEO33 + BR38, GEO50 + BR38 and GEO55 + BR38) with differing levels of genotype failures, we simulated 10%, 20% and 30% missing data proportions in each country using Dataset 3 and performed 10-fold cross-validations using the BALK classifier. The simulated genotyping failures had the greatest impact on the GEO33 barcode (Fig. 3 and accompanying source data in Supplementary Data 2, Supplementary Table 6). The pooled (country wide) median MCC for GEO33 dropped from 0.96 with no missing data to 0.89, 0.81 and 0.73 with 10%, 20% and 30% missing data respectively. The impact of missing data on the combined GEO33 + BR38 panel was lower, with pooled median MCCs dropping from 1.00 with no missing data to 0.98, 0.96 and 0.94 with 10%, 20% and 30% missing data respectively. In all other panels, the pooled median MCC dropped by ≥0.1 between the simulations with no (0%) versus 30% missing genotype calls: from 0.87 to 0.77 at BR38, 0.96 to 0.85 at GEO50, 0.98 to 0.89 at GEO55, and 1.00 to 0.98 at GEO50 + BR38 and GEO55 + BR38.

**Assessment of the candidate SNP panels in independent validation samples**. After exclusion of low-quality and suspected imported samples, a total of 142 samples (Independent Validation Dataset) that were not included in the training (i.e., not in Dataset 1, 2 or 3) were available to independently evaluate the performance of the candidate SNP panels with the trained classifiers. The Independent Validation Dataset comprised samples from each of 7 countries that were represented in the training dataset (Dataset 2). The geographic clustering patterns of the Independent Validation Dataset relative to the training dataset is illustrated in the neighbour-joining trees in Supplementary Fig. 3. The prediction performance of the samples in the Independent Validation Dataset at the SNP panels with the trained classifiers is presented in Table 2. The BR38 panel exhibited the lowest prediction accuracy, with a pooled (country wide) median MCC of 0.44. The GEO33 panel also exhibited generally low prediction accuracy (pooled median MCC = 0.64), but this was improved in the combined GEO33 + BR38 panel (pooled median MCC = 0.81). The GEO50, GEO55, GEO50 + BR38 and GEO55 + BR38 panels all exhibited generally high prediction accuracy with pooled median MCCs exceeding 0.80 (range 0.83-0.89). Figure 4 presents heat maps for each of the SNP panels illustrating the proportion of correct recalls for each country of origin (source data is provided in Supplementary Data 3). The heat maps demonstrate that, in all SNP panels, incorrect classifications generally reflected predictions to neighbouring countries, thus retaining regional geographic mapping accuracy.

## Discussion

The primary objective of the study was to develop molecular tools amenable to population-based surveillance frameworks that can be used to identify, and map imported *P. vivax* infections. Three new SNP panels (GEO barcodes) were identified with high country classification performance, that were able to distinguish imported *P. vivax* infections across a range of endemic scenarios. The most parsimonious panel, GEO33, exhibited high country classification when there was no missing data, and can be cost-effectively appended to the 38 bi-allelic, assayable Broad barcode SNPs (BR38) for improvement in predictive capacity in samples with moderate levels of missing data. The combined GEO33 + BR38 barcode generated robust country classification in most endemic areas, even when the proportion of missing data rose to 30%. However, the predictive capacity of the GEO33 + BR38 barcode between Cambodia and Vietnam was moderate, likely reflecting frequent human and associated *P. vivax* gene flow across the border between these two countries. The GEO50 and

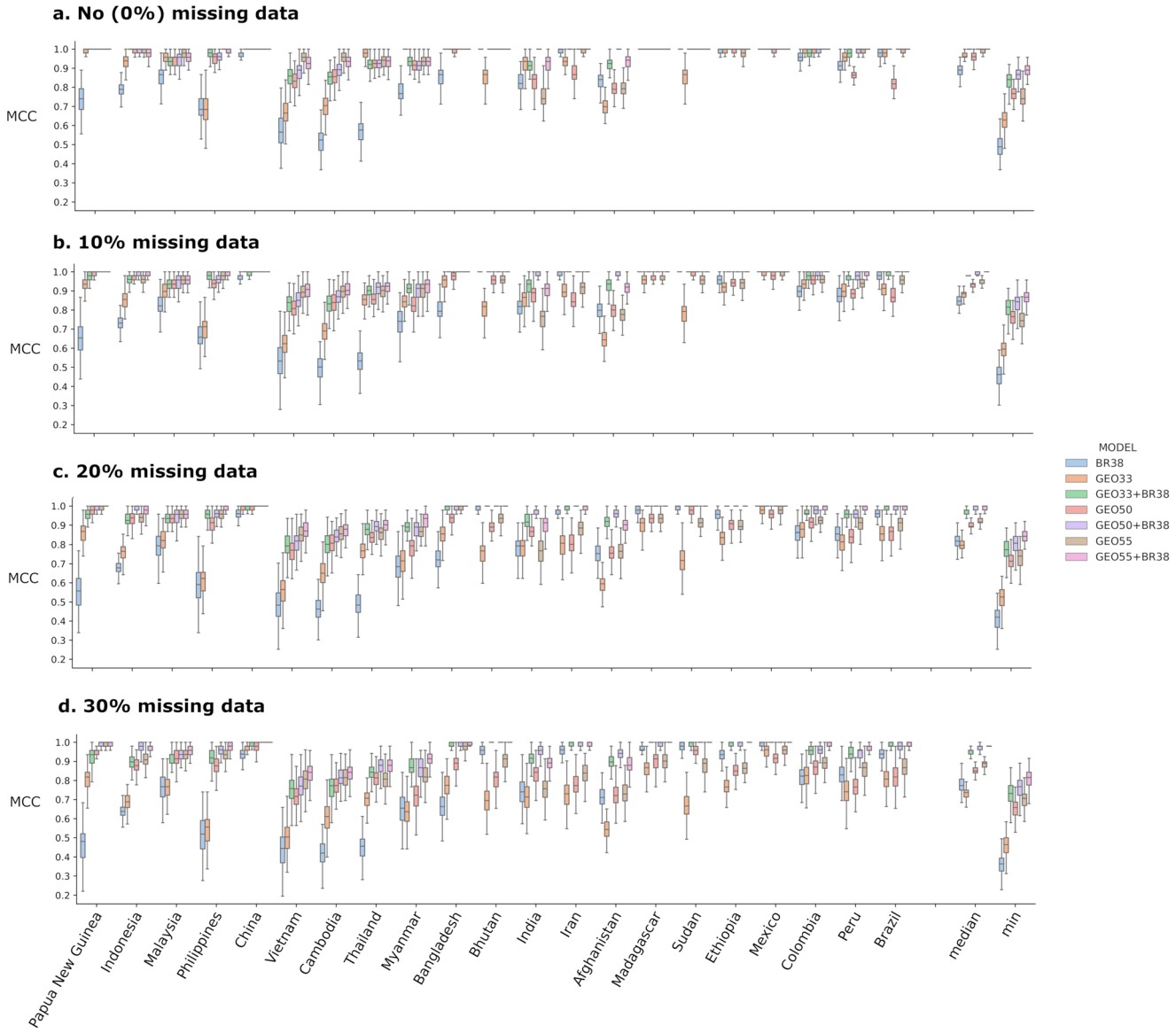

**Fig. 3 Comparison of country prediction performance between the SNP panels with simulated missing data.** MCC scores generated from 250 repeats with $n = 25$ biologically independent samples per country with no (0%) missing data (**a**) and simulating missing data (genotype fails) of 10% (**b**), 20% (**c**) and 30% (**d**); median and min reflect the respective summary statistics for the pooled MCC scores across all countries. Each bar presents the median, interquartile range and min and max MCC for the given country and model. With missing data, the combined BR38 and GEO panels (i.e., BR38 + GEO33, BR38 + GEO50 and BR38 + GEO55) demonstrated better results than the single panels in retaining prediction performance, likely owing to moderate levels of redundancy between some of the SNPs.

**Table 2 Summary of MCC scores in the independent validation dataset.**

| Country | N | BR38 | GEO33 | GEO33 + BR38 | GEO50 | GEO50 + BR38 | GEO55 | GEO55 + BR38 |
|---|---|---|---|---|---|---|---|---|
| Brazil | 7 | 0.742 | 0.551 | 0.814 | 0.655 | 0.742 | 0.808 | 0.881 |
| Cambodia | 65 | 0.412 | 0.635 | 0.674 | 0.903 | 0.833 | 0.860 | 0.890 |
| Colombia | 1 | 0.441 | 0.495 | 0.705 | 1.000 | 1.000 | 1.000 | 1.000 |
| Ethiopia | 18 | 0.902 | 0.902 | 0.968 | 1.000 | 1.000 | 0.907 | 1.000 |
| Peru | 14 | 0.715 | 0.921 | 0.921 | 0.842 | 0.854 | 0.897 | 0.921 |
| Thailand | 12 | 0.280 | 0.764 | 0.859 | 0.818 | 0.818 | 0.765 | 0.818 |
| Vietnam | 25 | 0.357 | 0.521 | 0.575 | 0.887 | 0.776 | 0.833 | 0.867 |
| Pooled median | 142 | 0.441 | 0.635 | 0.814 | 0.887 | 0.833 | 0.86 | 0.89 |
| Pooled min | 142 | 0.28 | 0.495 | 0.575 | 0.655 | 0.742 | 0.765 | 0.818 |
| Pooled Q1 | 142 | 0.3845 | 0.536 | 0.6895 | 0.83 | 0.797 | 0.8205 | 0.874 |

The analyses were based on $n = 142$ biologically independent samples.

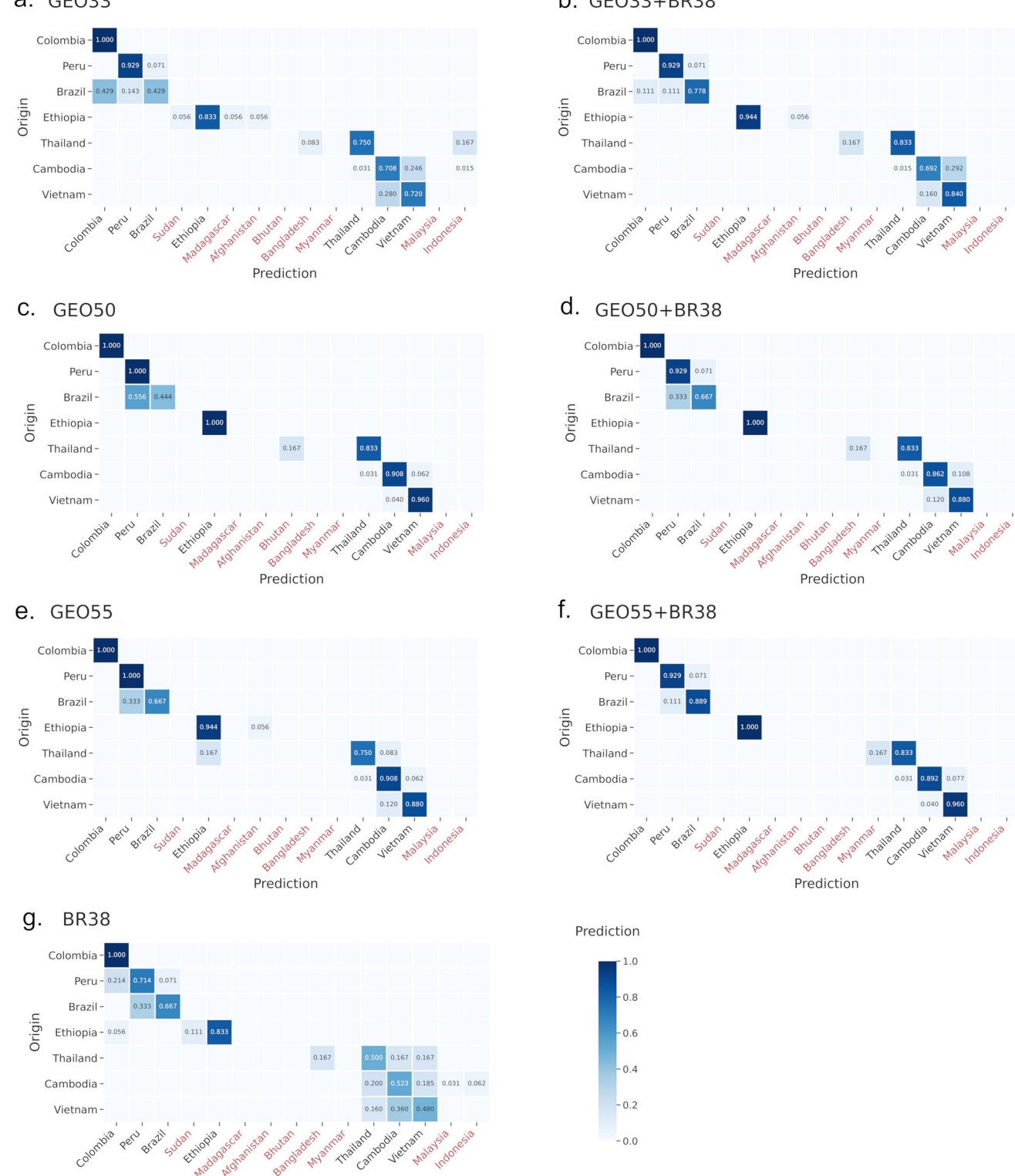

**Fig. 4 Heatmap illustrating country prediction accuracy at the BR38 and GEO barcodes in the Independent Validation Dataset.** Each plots presents the prediction performance of the given SNP panel (panels **a–g**) in the Independent Validation Dataset ($n = 142$ biologically independent samples) visualized as a heatmap showing the correlation between country of Origin and Prediction. Each cell is colour-coded to reflect the proportion of samples from the given country of Origin that were correctly assigned to the corresponding Prediction country. Colour-coding scaled from light blue (low proportion) to dark blue (high proportion). Only countries that were predicted by at least one of the SNP panels are presented, and Prediction countries that were not represented in the Independent Validation Set (i.e., not on the Origin axis) are labelled in red. Where samples' country of Origin did not directly match the country of Prediction, they generally mapped to neighboring countries (i.e., still within the correct regional geography). The BR38 panel exhibited lower prediction accuracy than the GEO and combined GEO + BR38 panels. Across the SNP panels, majority of incorrect predictions occurred between Cambodia, Vietnam and Thailand.

GEO55 panels achieved better resolution than the GEO33 + BR38 panel in these areas, and even greater characterization of parasite transmission across borders with high levels of gene flow may be possible with additional markers suited to an analysis of identity-by-descent[17]. In some geographic regions, where national borders have little or no impediment on parasite gene flow, even genome-wide data will not provide resolution of infections between neighboring countries: in these regions, country-level classification of infection origin may have limited utility. However, using genetic data to demonstrate that the parasites from different sides of the border form a single homogenous population may be helpful to strengthen the case for cross-country collaborative efforts to tackle vivax malaria. Furthermore, the tools described in this study can be adapted to characterize other population boundaries that may be of relevance to NMCPs. As the density of available genomic data on *P. vivax* increases, it may also be possible to use higher-resolution genetically defined infection boundaries for classification purposes.

The application and wider validation of the new GEO barcodes is underway, with Illumina amplicon-based sequencing assays already established by the Wellcome Sanger Institute malaria program for the 38-Broad barcode SNPs[13] and by collaborators at the Institute for Tropical Medicine, Antwerp, for GEO-33[18]. Further work will be needed to establish frameworks for implementation of parasite genotyping into the day-to-day activities of NMCPs: insights may be gained from the GenRe-Mekong framework, which has successfully implemented parasite genotyping into NMCP activities in several countries in the Greater Mekong Subregion for the purpose of tracking antimalarial drug resistance in *P. falciparum*[7]. The GenRe-Mekong framework currently focuses on conducting genotyping using the Illumina platform in centralized laboratories (such as national reference laboratories) with strong molecular biology expertise and equipment. However, assays for the geographic barcodes described in this study can be readily designed for other genotyping platforms such as the highly portable minION sequencers (Oxford Nanopore Technologies), which can theoretically be implemented in environments with minimal molecular laboratory equipment.

The analysis and interpretation of "real-world" genotyping data raises substantial challenges from low-quality samples such as those collected on dried blood spots. In anticipation of these needs we established a likelihood-based classifier framework with the capacity to deal with polyclonal infections as well as missing data. This framework has been integrated into the vivaxGEN-geo online platform (http://geo.vivaxgen.org), so that users can analyze and interpret their data without needing complex bioinformatics skills and avoiding the subjective visual inspection of neighbour-joining trees or principal component plots. Whilst the informatics tools implemented in vivaxGEN-geo are tailored to *P. vivax*, a similar approach can be adapted to other species. To facilitate wider application the source code is publicly available.

The variants in the GEO SNP panels are located in genes representing a range of functions, some of which may be unstable over time such as the variants in drug resistance-associated genes. These variants can readily be replaced with new variants as populations evolve. The rate at which allele frequencies change in a population will depend on various factors including the population size, extent of gene flow, and selection dynamics.

Although our dataset represents one of the most geographically diverse panels of *P. vivax* isolates currently available, with representation of all the major vivax-endemic regions, the predictive capacity of the derived tools is likely to be constrained by the geographic representation of the reference panel. The classifier cannot assign a prediction to a country that is not represented in the genetic reference panel, and countries that have a small or non-representative reference sample set may have limited classification accuracy. The limited representation from areas such as the Indian subcontinent is an important gap that needs to be filled. However, the reference panel has good representation of isolates from regions of public health relevance, including the epicenter of chloroquine-resistant *P. vivax* in Papua Indonesia, western Thailand and Myanmar, where a high frequency of *P. vivax* infections with mefloquine resistance-associated *MDR1* (*PVP01_1010900*) copy number variants have been reported, and Ethiopia, which comprises the largest reservoir of *P. vivax* in Africa and where infections that are able to invade duffy negative human red blood cells have been reported[19–24]. The strong representation of these areas in the genetic reference panel ensures that NMCPs can accurately identify when infections have been imported from these regions and effect appropriate case management responses. It is also important to acknowledge that the likelihood-based classifier framework is amenable to re-evaluation of the current maker sets as new genomic data become available, facilitating iterative development of refined SNP panels. As the reference panel expands with increasing data generated at the barcode SNPs, the accuracy of the likelihood-based classifications will improve.

The likelihood-based classifier framework has been designed to allow geographic predictions to be assigned to polyclonal infections carrying two or more clones, as are common in high endemicity regions; these infections are commonly omitted from population genetic analyses. However, it should be acknowledged that the classifier does not attempt to phase individual clones, rather the infection is analyzed as a composite, yielding a single prediction of most likely origin. Nonetheless, it is important to note that, by design, the GEO panels selected by the framework should exhibit low within-country diversity (with diversity rather being between countries). Polyclonal infections deriving from a single country should therefore exhibit a low frequency of heterozygote positions at the selected GEO barcodes. In cases where a combination of clones deriving from different countries are present within a single infection, yielding many heterozygote positions, the classifier will be constrained in its ability to detect country of origin and a low confidence in the prediction will accordingly be assigned. Future developments that combine GEO markers with high-resolution finger-printing markers such as microhaplotypes may enable polyclonal infections to be phased and subsequently analyzed for geographic origin.

As well as new geographic markers, future iterations of the SNP barcode are being developed to address other use cases. These will include markers of drug resistant *P. vivax* as well as markers to characterize recurrent infections, that will support the interpretation of clinical trials, epidemiological cohorts and parasite surveillance (see microhaplotype description in[8]). Whilst the geographic origin of a *P. vivax* infection can provide some insights into a parasite's likely relapse periodicity, the risks and frequency of recurrent infections are influenced by a range of factors including transmission intensity, hypnozoite burden and host immunity, which confound correlation between parasite genotype and an individual's risk of relapse[4,25].

In 2017, up to 100% of all confirmed malaria cases in 17 malaria-endemic countries in the Asia-Pacific region, the Middle East and the Americas, where *P. vivax* infections predominate, were reported as being imported infections[1]. In these countries national malaria control programs can utilize information derived from our molecular tools to assess the efficacy of ongoing interventions in reducing local transmission. One of the key requirements by the World Health Organization for certifying malaria elimination is demonstration that all malaria cases detected in-country over at least three consecutive years were imported. Our genotyping approach has potential to identify imported infections

thus reducing ambiguity in elimination certification. For this purpose, countries approaching elimination will need to maintain archival samples for future molecular comparisons against putatively imported cases.

The molecular *P. vivax* geographic classification tools presented are designed to empower users in malaria-endemic countries to compare local genotyping data with globally available datasets. Amplicon-based sequencing of geographic barcodes will be combined with other surveillance markers at central laboratories in endemic partner countries of the Asia Pacific Malaria Elimination Network (www.apmen.org). The data generated from these centers will inform researchers, National Malaria Control Programs and other key stakeholders on the incidence, epidemiology and key reservoirs of imported malaria and, in doing so, help to target resources to where they are needed most.

## Methods

**Overview of data analysis methods.** The project aimed to generate two major outputs: a new framework to identify *P. vivax* geographic barcodes (i.e., marker selection) and an online, open-access informatics platform for end-users to analyze data generated using the barcode. A flow diagram outlining the steps involved in identifying *P. vivax* geographic barcodes is provided in Fig. 1. The process involved three key steps: 1) data preparation to produce a dataset with the optimal balance of number of samples and SNPs and with no missing data (i.e. no genotype fails), 2) SNP selection using Decision Tree and HFST approaches to obtain candidate SNP panels suitable for the classifier developed in this study (a Bi-Allele Likelihood, BALK classifier) and, and 3) comparative evaluation of the candidate SNP panels, assessment of the impact of missing data (i.e., genotype fails), and assessment of the prediction accuracy with an independent dataset. An online, open-access informatics platform was then developed and equipped with BALK classifiers trained against the candidate SNP panels. A more comprehensive description of the methods is provided in the Supplementary Methods.

**Data set.** The study used genomic data on *P. vivax* derived from the Malaria Genomic Epidemiology (MalariaGEN) *P. vivax* Genome Variation Project release 4 (Pv4), which has recently been published as an open dataset[26]. The Pv4 open data set comprises genomes from 26 countries. At the time of conducting our analysis (i.e., prior to the Pv4 open access release), a dataset comprising 1873 (of the 1895 samples described in the release) samples was available for our study. For the analysis in this study, the dataset was divided into two parts, a training dataset, and a validation dataset. The validation set consisted of isolates from 7 countries (Brazil, Cambodia, Colombia, Ethiopia, Peru, Thailand, and Vietnam) derived from a clinical trial conducted by GlaxoSmithKline (GSK)[26]. All remaining isolates were included in the training dataset, which comprised representation of all the countries in the validation set. The GSK samples were selected for independent validation owing to convenience as the samples from this study were sequenced later than the other studies and, hence, the data was made available later.

**Data preparation.** An overview of the data preparation steps is outlined in section a) of the flowchart presented in Fig. 1. Briefly, the training dataset was filtered to exclude recurrent infections and samples from countries represented by less than 4 independent *P. vivax* genomes, resulting in an initial dataset comprising 1,348 samples from 21 countries (Supplementary Table 1, Supplementary Fig. 4). With this initial dataset, from the initial 2,671,112 variants discovered in the MalariaGEN Pv4 project[26], we derived a set of 662,641 high-quality bi-allelic SNPs with VQSLOD score > 0, minimum depth of 1 and minimum Minor Allele Count (MAC) of 2 to produce Dataset 0. The restriction to bi-allelic SNPs is a standard approach undertaken in malaria population genomics to simplify downstream computations and does not impose constraints on the analysis of polyclonal infections, which are still detectable through the composite of allelic variants across the respective SNPs (see[27–29]). Individual genotype calls were defined as heterozygotes based on an arbitrary threshold of a minor allele ratio > 0.1 and a minimum of 2 reads for each allele; all other genotype calls were defined as homozygous for the major allele. Dataset 0 was further filtered to exclude non-independent samples, defined arbitrarily as isolate pairs with genetic distance less than 0.001, resulting in 1,227 samples with 662,641 SNPs, denoted as Dataset 1. Dataset 1 was then subjected to iterative data quality filtering to derive the best representative number of samples and informative SNPs without any genotype missingness by iteratively removing samples with higher missingness and calculating the number of informative SNPs (defined as SNPs with MAC >= 2), from the remaining samples. Based on the plot of the result of this data quality filtering (Supplementary Fig. 1), we identified 826 samples and 229,317 SNPs to be included in Dataset 2. The isolates in Dataset 2 were initially assigned to country based on the available metadata, which was further evaluated using 1) country-level prediction using the BALK classifier against all 229,317 SNPs and 2) manual confirmation by

constructing a neighbor-joining tree based on genetic distance. Isolates whose country assignment differed from the prediction result and that were not in the same country cluster as observed manually from the neighbor-joining tree were considered suspected imported infections and removed from the dataset to produce Dataset 3, comprising 799 samples and 229,317 SNPs. For comparative assessment of candidate SNP panels, a new dataset (Dataset 4) was produced which comprised the samples in Dataset 3, but only the SNPs selected by the consecutive SNP selection process (we refer to these SNP panels as GEO barcodes) and 38 assayable SNPs from a commonly used 42-SNP *P. vivax* barcode developed by the Broad institute[12]. The SNP panel comprising the 38 assayable Broad Institute barcode SNPs is referred to as BR38. The BR38 SNP panel was integrated in the study for evaluation on its own as well as in combination with the newly selected GEO SNP panels as it has been implemented in several countries.

A similar filtering process was applied to the validation set. All recurrent infections were removed, and the SNP positions were filtered to only include the 229,317 SNPs defined in the training Dataset 4. Any remaining non-independent samples were then removed using the same 0.001 threshold of genetic distance, using a similar procedure to that described for the training set. Country-level assignment was assessed using the same trained BALK classifier as the training set, and a neighbor-joining tree was constructed by combining with Dataset 3 for manual confirmation. After the various filters, a set of 142 samples remained in the validation set. Supplementary Fig. 2 presents the neighbor-joining tree of Dataset 3 combined with the 142 validation samples at the 229,317 SNPs. Further SNP filtering to only include the BR38 panel and newly selected GEO SNPs were performed to produce the Independent Validation Dataset. More detailed information on the data preparation methods is available in the Supplementary Methods.

**Bi-Allele likelihood classifier.** Our study required the development of flexible methods to classify *P. vivax* infections/genetic data by country. For this purpose, we required a classifier with the following properties: 1) capable of evaluating existing SNP panels, 2) amenable to new SNP additions to accommodate new countries or genetic shifts over time, 3) able to classify data inputs containing genotype fails and bi-allelic heterozygous genotype calls arising from polyclonal infections, and 4) able to provide confidence values of prediction. We identified the Naive Bayes classifier as having the properties that cater to the above requirements after application of several modifications. We derived a Bi-Allele Likelihood (BALK) classifier from Bernoulli Naive Bayes with modification by replacing the likelihood equation of its classification rule from the Bernoulli probability distribution to a binomial $N = 2$ distribution to handle the heterozygous calls and setting the prior probability to a uniform distribution, making the classifier only depend on the likelihood of the SNP data. The BALK classification rule is presented in equation 1.

$$\max Pr(C|X) \sim Pr(X|C) = \prod_{i}^{n} p_i^{x_i} \cdot (1 - p_i)^{(2 - x_i)} \qquad (1)$$

Where $X$ is the SNP data set of a sample, $C$ is a group (or a country), $x_i$ is the number of alternate alleles at position $i$ and $p_i$ is the frequency of the alternate allele at position $i$ of country $C$ counted as diploid samples. A more comprehensive description of the development of BALK classifier is available as Supplementary Methods.

**Candidate SNP selection.** Our objective was to identify the most parsimonious SNP panels for country-level classification, aiming for less than 60 SNPs in these panels; this threshold for the new SNP panels was based on several considerations. In accordance with the multiplexing features of the Illumina platform and considering primer, library preparation and sequencing costs, as well as the practical challenges of preparing PCR pools across large numbers of primers, we identified a maximum of 100 SNPs in total (across the new SNP panels and previously described Broad barcode i.e., BR38) as a feasible threshold for a geographic barcode for *P. vivax*.

An overview of the candidate SNP selection steps is outlined in section b) of the flowchart presented in Fig. 1. Optimal SNPs for country classification were selected using the following approaches: DecisionTree, HFST-0.90 and HFST-0.95 (HFST with Fst threshold of 0.9 and 0.95 respectively), which are detailed in the Supplementary Methods. Briefly, for the DecisionTree (DT) approach, Dataset 3 was subjected to a DT implementation of the Python sklearn library. The SNP set selected by DT was then reassessed with the BALK classifier in the training set using country-level MCC (Matthew Correlation Coefficient) scores as well as pooled (cross-country) median and minimum MCC scores. The MCC provides a measure of the quality of the classifications, ranging from -1 (total disagreement) to 1 (perfect prediction)[30]. For the HFST (Hierarchical FST) approach, as a bifurcating tree guide, a neighbor-joining population tree was constructed based on Nei's net average population genetic distance matrix and then re-rooted at the midpoint (Supplementary Fig. 5). The HFST approach entailed traversing across the bifurcating guide tree and randomly selecting the SNPs with FST higher than a certain threshold between the two populations represented by the two nodes of the branch. If none of the SNPs were above the threshold during guide tree traversal, the DT method was employed to obtain additional SNPs to separate the two nodes of the branch. As with the DT approach, country-level MCC scores and pooled

(cross-country) median and minimum MCC scores of each of selected SNP set were calculated using the BALK classifier trained against the selected SNPs with Dataset 3.

For each approach, Dataset 3 was used for both training and testing set in 500 repeats to obtain 500 SNP sets. The top-25 SNP sets from the 500 SNP sets, ranked based on the average of their minimum MCC and median MCC scores over country-level MCC scores, were collected and subjected to the 500 repeats of stratified 10-fold cross-validation to avoid over-fitting each SNP set by re-ranking based on their average minimum MCC and median MCC scores to derive the best SNP set for each approach.

**Comparative assessment of candidate SNP panels**. An overview of the steps involved in the comparative evaluation of the SNP panels is outlined in section c) of the flowchart presented in Fig. 1. To compare the Broad SNP panel to the three new candidate SNP panels identified by DT, HFST-0.90, and HFST-0.95 approach a 500 repeat, stratified 10-fold cross-validation was undertaken on each SNP panel using Dataset 3.

Additionally, to assess the durability of prediction performance of the candidate SNP panels with different levels of missing data (analogous to genotyping failures), simulations were run after removing genotype data randomly. The BALK classifier was trained against the candidate SNP panels using all samples. For each country, 25 samples were sampled randomly with replacement and genotype calls were removed from the SNP sets in 10%, 20% and 30% proportions. The random samples were then subjected to the trained classifier. This process was run in 250 repeats and the MCC score of the prediction for each country was reported.

To evaluate the performance of the candidate SNP panels with new sample sets (as opposed to using the re-sampling technique of the cross-validation strategy), the trained BALK classifiers were run on the Independent Validation Dataset and MCC scores reported for each country.

**Web-based data analysis and sharing platform for end-users**. To establish accessible informatics tools for end users, an online platform was created incorporating data classification tools for determining the most likely country of origin of a sample using genetic data derived from different barcodes. Existing source code, developed for a microsatellite-based *P. vivax* data sharing and analysis platform[31], was modified to create a new web-based platform (vivaxGEN-geo), to collate SNP data generated at the geographic barcode. This approach was chosen owing to the ability to i) incorporate manual SNP sets allowing incremental improvements of the barcode in future, ii) evaluate barcodes with incomplete data owing to genotyping failures, and iii) evaluate heterozygous genotype calls, which reflect polyclonal infections. For optimal accuracy, the BALK classifier provided on the online platform has been trained with 941 samples, comprising Dataset 2 ($N = 799$) plus the Independent Validation Dataset ($N = 142$). The classifier tool reports the three highest likelihoods for country of origin and their associated probabilities. The classifier tool reports the three highest likelihoods for country of origin and their associated probabilities. The probabilities were computed using the isotonic method as implemented in CalibratedClassfierCV of sklearn library, with stratified 4-fold cross-validation for the calibration dataset. The web platform can receive the input data in string-based barcode representation, column-based tab-delimited text files, and VCF files.

**Ethics**. All samples were collected with written informed consent from patients, or their legal guardians as detailed in the Malaria Genomic Epidemiology (Malaria-GEN) *P. vivax* Genome Variation Project release 4 data note[26].

**Reporting summary**. Further information on research design is available in the Nature Portfolio Reporting Summary linked to this article.

## Data availability

The study used genomic data from the MalariaGEN *P. vivax* Genome Variation Project release 4 (Pv4)[26]. VCF and zarr format files containing the genotype calls used in the study are available open access on the MalariaGEN data resource page at https://www.malariagen.net/resource/30[26].

## Code availability

All custom, in-house scripts used for data filtering, analyses and visualization are available from https://github.com/vivaxgen/geo. The VivaxGEN-geo web service is accessible at http://geo.vivaxgen.org/. In addition to the new geographic SNP panels described in this study, vivaxGEN-geo provides classification of other SNP panels including a published Vietnamese barcode (VN40)[18].

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

## Acknowledgements

We would like to thank the patients who contributed their samples to the study, and the health workers and field teams who assisted with the sample collections. We also thank the staff of the Wellcome Sanger Institute Sample Logistics, Sequencing, and Informatics facilities for their contributions. For the purpose of open access, the author has applied a CC BY public copyright license to any Author Accepted Manuscript version arising from this submission. This research was funded in part by the Wellcome Trust (Senior Fellowship in Clinical Science awarded to R.N.P., 200909). The research was also funded in part by the Australian Department of Foreign Affairs and Trade (TDCRRI 72904), the Australian National Health and Medical Research Council (NHMRC) (APP2001083 awarded to S.A.), and the Bill and Melinda Gates Foundation (OPP1164105). H.T. was supported by a Charles Darwin University International PhD Scholarship (CDIPS). The patient sampling and metadata collection was funded by the Asia-Pacific Malaria Elimination Network (108-07), the Malaysian Ministry of Health (BP00500420), and the NHMRC (1037304 and 1045156; Fellowships to N.M.A. [1042072 and 1135820], B.E.B. [1088738] and M.J.G. [1074795]). M.J.G was also supported by a 'Hot North' Earth Career Fellowship (1131932). M.U.F is supported by a senior researcher scholarship from the Conselho Nacional de Desenvolvimento Científico e Tecnológico (CNPq), Brazil. The whole genome sequencing component of the study was supported by grants from the Medical Research Council and UK Department for International Development (M006212) and the Wellcome Trust (204911) awarded to D.P.K., and a Wellcome Trust grant (206194/Z/17/Z) awarded to D.P.K. and J.C.R. This work was supported by the Australian Centre for Research Excellence on Malaria Elimination (ACREME), funded by the NHMRC (APP 1134989).

## Author contributions

S.A., H.T., R.A., R.D.P., and R.N.P. conceived and designed the study, and wrote the first draft of the manuscript. S.A., H.T., R.A., R.D.P., and E.S. conducted data analysis. R.N., L.T., J.M., Z.P., D.E.F., T.M.L-M., L.M.M., A.T-C., M.J.G., B.B., T.W., N.M.A., S.G., B.P., A.Aseffa, A.Assefa, A.G.R., N.H.C., T.T.T., M.S.A., W.A.K., B.L., K.T., S.W., Y.H., I.A., Y.L., Q.G., K.S., M.U.F., M.L., A.B., I.M., M.V.G.L., A.L-C., S.K., C.L., R.M., D.Y., D.B.P., F.E.J.E., C.S.C., I.D.V., C.N-L., M.F.V., J.A.G., G.K., N.J.W., and F.N. contributed critical field-based malaria collections and metadata. D.P.K., J.C.R., R.A., R.D.P., E.D., S.G., V.S., O.M., and A.M. contributed sequencing, data production and informatic support.

## Competing interests

The authors declare no competing interests.

## Additional information

[1]Global and Tropical Health Division, Menzies School of Health Research and Charles Darwin University, Darwin, NT, Australia. [2]Eijkman Institute for Molecular Biology, Jakarta, Indonesia. [3]Wellcome Sanger Institute, Wellcome Genome Campus, Cambridge, UK. [4]Exeins Health Initiative, Jakarta, Indonesia. [5]International Training and Medical Research Center (CIDEIM), Cali, Colombia. [6]Departamento de Microbiología, Universidad del Valle, Cali, Colombia. [7]Universidad Icesi, Cali, Colombia. [8]Malaria Group, Universidad de Antioquia, Medellin, Colombia. [9]Infectious Diseases Society Sabah-Menzies School of Health Research Clinical Research Unit, Kota Kinabalu, Sabah, Malaysia. [10]Clinical Research Centre, Queen Elizabeth Hospital, Sabah, Malaysia. [11]College of Natural Sciences, Addis Ababa University, Addis Ababa, Ethiopia. [12]Armauer Hansen Research Institute, Addis Ababa, Ethiopia. [13]Ethiopian Public Health Institute, Addis Ababa, Ethiopia. [14]Mahidol-Oxford Tropical Medicine Research Unit, Mahidol University, Bangkok, Thailand. [15]Nangarhar Medical Faculty, Nangarhar University, Ministry of Higher Education, Jalalabad, Afghanistan. [16]Oxford University Clinical Research Unit, Hospital for Tropical Diseases, Ho Chi Minh City, Vietnam. [17]Infectious Diseases Division, International Centre for Diarrheal Diseases Research, Dhaka, Bangladesh. [18]Royal Center for Disease Control, Department of Public Health, Ministry of Health, Thimphu, Bhutan. [19]Infectious and Tropical Diseases Research Center, Hormozgan University of Medical Sciences, Bandar Abbas, Hormozgan Province, Iran. [20]Faculty of Medicine, University of Khartoum, Khartoum, Sudan. [21]National Health Commission Key Laboratory of Parasitic Disease Control and Prevention, Jiangsu Provincial Key Laboratory on Parasite and Vector Control Technology, Jiangsu Institute of Parasitic Diseases, Wuxi, China. [22]School of Public Health, Nanjing Medical University, Nanjing, China. [23]Shoklo Malaria Research Unit, Faculty of Tropical Medicine, Mahidol University, Mae Sot, Thailand. [24]Department of Parasitology, Institute of Biomedical Sciences, University of São Paulo, São Paulo, Brazil. [25]Global Health and Tropical Medicine, Institute of Hygiene and Tropical Medicine, NOVA University of Lisbon, Lisbon, Portugal. [26]Papua New Guinea Institute of Medical Research, Madang, Papua New Guinea. [27]Deakin University, Victoria, Australia. [28]Population Health and Immunity Division, The Walter and Eliza Hall Institute of Medical Research, Victoria, Australia. [29]Department of Medical Biology, The University of Melbourne, Victoria, Australia. [30]Department of Parasites and Insect Vectors, Institut Pasteur, Paris, France. [31]Fundação de Medicina Tropical, Manaus, Brazil. [32]Fundação Oswaldo Cruz, Manguinhos, Rio de Janeiro, Brazil. [33]Universidad Peruana Cayetano Heredia, Lima, Peru. [34]Mahidol University, Bangkok, Thailand. [35]Armed Forces Research Institute of Medical Sciences, Bangkok, Thailand. [36]University of Gondar, Gondar, Ethiopia. [37]Jimma University, Jimma, Ethiopia. [38]Centro de Pesquisa em Medicina Tropical, Porto Velho, Brazil. [39]Research Institute for Tropical Medicine, Manilla, Philippines. [40]Umphang Hospital, Tak, Thailand. [41]Centro de Investigaciones Clinicas, Cali, Colombia. [42]GlaxoSmithKline, Brentford, UK. [43]Cambridge Institute for Medical Research, School of Clinical Medicine, University of Cambridge, Cambridge, UK. [44]Centre for Tropical Medicine and Global Health, Nuffield Department of Medicine, University of Oxford, Oxford, UK. ✉email: Sarah.Auburn@Menzies.edu.au

