## [Peer Review File · Communications Biology]

Reviewer #1 (Remarks to the Author):

In the manuscript, the authors presents a classifier of *P. vivax* genome to their origin country based on SNP panels. They compared the performance of an existing panel, a newly designed panel from training genome data and select for the best combination using HFST and DT methods. I think the topic of the manuscript, and the publicly available web-interface are useful for the local epi control centers. However, the description of the methodology is poorly written, and lacks reference to the theory of the classifier, population genetics, and justifications from statistics. I don't think it can be published in the current state.

There has been a very rich literature on how to infer distinct populations from multi-locus genotype data, and assign individuals to populations, such as STRUCTURE developed by Pritchard, Stephens & Donnelly (2000). Pritchard et al. (2000) also used a likelihood based model, except that they also treated the number of separate populations (k in this manuscript), SNP frequency per population (p_{ki} in this populations) as unknowns that need to be estimated/inferred. In this manuscript, the authors assumed that each country necessarily form a separate population of *vivax*, while in fact, neighbouring countries could represent one population, or there could be several distinct populations within one large country. The most problematic assumption of the LK classifier is to assume P_{ki} is known, where the authors didn't describe how they obtained these values. If the authors just directly use the SNP frequency from the included samples, then there could be very strong bias from countries with low sample sizes (which is the case for majority of the countries included that have less than 10 samples in total).

The part that describes candidate SNP selection (from Line 257) is very quite impossible to understand. HFST and DT are not described clearly, with no references either, or any supplementary text. It's also unclear how the 10-fold cross-validation is performed. While assessing the performance of the SNP panels, it is also unclear to me whether the low uncertainty of assignment for a lot of the countries can be attributed to the good performance of the model, or simply because they are represented by a very small number of samples. Similarly, the country assignment errors are higher for Vietnam and Cambodia could be attributed by the fact that the countries have constant gene flow, and some of the genomes are recombinants of local and imported genomes from sometime ago.

Since human movements are frequent between borders, it is also important to estimate the timeframe for the current SNP panels to be useful. For example, when certain countries are close to elimination, and majority of their cases are imported, it is hard to differentiate which genomes are local and which ones are imported after a while. The frequencies of these SNPs are also subject to drastic changes when interventions are implemented or human movements are frequent.

Minor suggestions:

- 1) how many strains are there in polyclonal infections? How are the authors certain that there are only two strains in these samples?
- 2) sub labels need to be added on Figure 1 and referenced in the methods section where each part is described. Currently, it's very hard to know the correspondence between the flowchart and the section in the text.
- 3) Axes need to be labeled in Supplementary Figure 1
- 4) Line 84, "simulations" mentioned in the manuscript multiple times. However, it's unclear what kind of simulations are run. Random removal of genotypes in an empirical sample cannot be referred to as simulations.

Reviewer #2 (Remarks to the Author):

The authors have used an innovative machine learning approach to select SNPs that can discriminate

the country origin of *P. vivax* infections. These SNP barcodes are increasingly important in resolving sources and sinks of infections and hence useful for countries that are moving towards elimination. It is commendable that the work has also led to the development of an openly available tool for use by non experts in better understanding the sources of infections. The translation of the barcode and tool across the global *P. vivax* endemic population will depend on access to data generation facilities, in this case deep amplicon sequencing. It should be noted that deep sequencing technologies remain inaccessible to most national public health facilities and the malaria control programs. This does not minimise the validity and future usefulness of the tools developed here.

There are a number of issues that could be addressed to further improve the paper.

1. The study takes opportunity of freely available genome data and therefore has no control in the design of the original studies that generated the data. In this case, it should be noted that the sampling cannot be generalised to represent a country as the samples could be geographically biased even within country besides being designed to access a specific population sub-group such as clinical cases for therapeutic efficacy studies. Within country variation can be palpable in some settings such as Ethiopia where gene flow between regions are subject to restrictions due to the geography of the population.

2. The census numbers for some populations were small and as stated above can hardly be representative of the country of origin. Was the effect of this small numbers simulated

3. The GSK studies were selected for validation. These were from some of the countries included in the training set. Hence the samples are not that independent, though the source studies are independent. Will an iteration through several random subsets from different studies and source populations not increase the confidence and generalisation of the selected markers.

4. The authors identified three new sets of markers upon the already existing 38 barcode from Broad Institute. The authors lean towards a 71 marker barcode when a new 33 SNPs are combined with the existing 38 and this seem to significantly improve the level of discrimination provide by either. However, the 50 and 55 barcodes perform comparably with these even with poor quality samples. Why was all combinations of set pairs not tested as for the 33 plus 38. I could not determine if each of the 4 sets were unique or there are overlapping SNPs in these different sets. Can there be a consensus set from combining all sets.

Response to reviewers

*Note, line number given in our responses refer to the clean manuscript (i.e., without tracks or track changes version with simple markup viewing)

Reviewer #1 (Remarks to the Author):

- 1. In the manuscript, the authors presents a classifier of *P. vivax* genome to their origin country based on SNP panels. They compared the performance of an existing panel, a newly designed panel from training genome data and select for the best combination using HFST and DT methods. I think the topic of the manuscript, and the publicly available web-interface are useful for the local epi control centers. However, the description of the methodology is poorly written, and lacks reference to the theory of the classifier, population genetics, and justifications from statistics. I don't think it can be published in the current state.***

We thank the reviewer for their helpful feedback, particularly with respect to machine learning. We had deliberately tried to make the manuscript accessible to potential users of the tools such as policy makers and malaria control program officers, many of whom have limited genetics knowledge. However, we acknowledge the reviewer's request for additional details on the genetics methods to inform readers with a keen interest in these methods. To avoid making the main text of the manuscript inaccessible to readers who have limited genetics or machine learning knowledge, we have provided further details on the machine learning methods as a new Supplementary Materials document (please find in the revision).

- 2. There has been a very rich literature on how to infer distinct populations from multi-locus genotype data, and assign individuals to populations, such as STRUCTURE developed by Pritchard, Stephens & Donnelly (2000). Pritchard et al. (2000) also used a likelihood based model, except that they also treated the number of separate populations (k in this manuscript), SNP frequency per population (p_{ki} in this populations) as unknowns that need to be estimated/inferred. In this manuscript, the authors assumed that each country necessarily form a separate population of *vivax*, while in fact, neighbouring countries could represent one population, or there could be several distinct populations within one large country. The most problematic assumption of the LK classifier is to assume P_{ki} is known, where the authors didn't describe how they obtained these values. If the authors just directly use the SNP frequency from the included samples, then there could be very strong bias from countries with low sample sizes (which is the case for majority of the countries included that have less than 10 samples in total).***

Our rationale for classifying by national boundaries in the current iteration of vivaxGEN-GEO reflects the World Health Organization (WHO) requirement for countries to report on cases as local or imported at the country administrative level in order to receive certification of malaria-free status. For this very specific use case, assessment of P_{ki} within-country was not informative, rather, we needed to force the selection of SNPs that are homologous across a given country. However, we agree with the reviewer that some countries won't form a separate population (i.e., won't follow national boundaries). We had

previously raised this point in the discussion section of our manuscript (where we describe the challenge of extensive gene flow across the Cambodia border with Vietnam), but have emphasized this further in lines 409-422 to ensure there is no confusion around this point.

An important point to note is that **our likelihood classifier tools can easily be adapted to classify infections by other boundaries**, such as regional boundaries or indeed genetically defined boundaries. More dense genetic information is needed to construct useful genetic boundaries, but this may be possible in future as the proportion of the vivax-endemic map for which we have genomic data available expands. In terms of regional classification, we have provided extra results in the rebuttal letter to illustrate the high classification potential at this administrative level. We refrained from adding the regional-level classifications to the manuscript as we were concerned that this might over-complicate the methodology (as the reviewer requested in an earlier comment, we have attempted to keep the methods write up as simple and clear as possible), but we can add these details to the study if the reviewer(s) request it.

Assessment of genetically defined regional-level geographic resolution

*Although the primary objective of the marker selection for the current study was to classify infections at the country level, as observed by the evidently high gene flow between countries such as Cambodia and Vietnam, classification at this administrative level may not be informative in some geographic regions. Using neighbour-joining methods to determine **genetic clustering patterns**, a recently published data note including the *P. vivax* genomes analyzed here described **8 regional-level populations** reflecting South America, Africa, Western Asia, West Southeast Asia, East Asia, East Southeast Asia, Maritime Southeast Asia, and Oceania (<https://wellcomeopenresearch.org/articles/7-136>). We assessed the potential of the 38-SNP Broad barcode (BR38) and the three new SNP panels (GEO barcodes) to classify the seven genetically defined regional groups. The regional level classification performance of i) BR38, ii) GEO33, iii) GEO33+BR38, iv) GEO50, v) GEO50+BR38, vi) GEO55, and vii) GEO55+BR38 was analyzed using a 500-repeat, 10-fold cross-validation using the LK classifier on the samples in Dataset 3. The results of the regional-level evaluations in Dataset 3 are illustrated in the **Figure A** below. The median MCC of the GEO33 SNP panel exceeds 0.8 in all geographic regions, while the median MCCs of the GEO50 and GEO55 SNP panels exceed 0.9 in all regions. **These results demonstrate the potential of the tools developed in this study to be readily modified to meet the requirements of the user.***

Figure A. Regional level prediction performance at the SNP panels. The boxplots present the MCC scores from 500 repeats with stratified 10-fold cross validation for each SNP set in the training dataset ($n=799$ samples). Regional labels are provided on the y-axis: AF (Africa), EAS (East Asia), ESEA (East Southeast Asia), MSEA (Maritime Southeast Asia), OCE (Oceania), SAM (South America), WAS (West Asia) and WSEA (West Southeast Asia). Median and min reflect the respective summary statistics for the pooled MCC scores across all countries.

We had previously acknowledged in the discussion that sample size is an issue in the current data set but have further emphasized this to ensure clarity in lines 450-467. **This framework represents a first building block, to which further data from other countries will be added**, continually enhancing the accuracy of the tools. As we mention in the revised discussion, the marker set is not expected to be static, rather the marker selection and evaluation framework as well as the genotyping platform have all been carefully designed/chosen to be amenable to future additions and enhancements.

- 3. The part that describes candidate SNP selection (from Line 257) is very quite impossible to understand. HFST and DT are not described clearly, with no references either, or any supplementary text. It's also unclear how the 10-fold cross-validation is performed. While assessing the performance of the SNP panels, it is also unclear to me whether the low uncertainty of assignment for a lot of the countries can be attributed to the good performance of the model, or simply because they are represented by a very small number of samples. Similarly, the country assignment errors are higher for Vietnam and Cambodia could be attributed by the fact that the countries have constant gene flow, and some of the genomes are recombinants of local and imported genomes from sometime ago.*

We apologize for the challenges in interpreting the likelihood-classifier methods. We have designated an entire **Supplementary Methods section** (pages 5-23 of the Supplementary Information) to clarify the methodology applied. We hope this meets the reviewer's requirements and are happy to further clarify or provide any further details if needed.

We agree with the reviewer that the country assignment errors for Cambodia and Vietnam are likely due to gene flow across the borders. We have openly described this in the discussion, lines 409-422. We also show this with the heat plot from the independent validation data set (Figure 4). As described above, our framework is adaptable to other boundaries such as regional and genetically defined boundaries.

- 4. Since human movements are frequent between borders, it is also important to estimate the timeframe for the current SNP panels to be useful. For example, when certain countries are close to elimination, and majority of their cases are imported, it is hard to differentiate which genomes are local and which ones are imported after a while. The frequencies of these SNPs are also subject to drastic changes when interventions are implemented or human movements are frequent.***

This is a good point. We acknowledge this and, indeed, had previously commented on this in the discussion, but we have emphasized this further for clarity (lines 445-449). As mentioned earlier, the marker selection and evaluation framework as well as the genotyping platform are all amenable to enhancements as new data is provided. Regarding how frequently these markers might need to change, it will depend on the epidemiological setting in question. To gauge this in our data set, we assessed temporal changes over a 13-year period using samples from one of our largest country sets, Thailand, where vivax malaria has declined steadily over the past decade. To avoid over-complicating the manuscript, we did not provide the results in the manuscript but have presented them below (blue text) for the reviewer. We can provide them in the manuscript if the reviewers request. We should also note that despite gene flow and recombination, assessment of global *P. vivax* populations shows genetic structure that is remarkably aligned with geography (see <https://wellcomeopenresearch.org/articles/7-136>). Furthermore, in areas where cross-border human movement is frequent, one does not have to be restricted to national boundaries if regional boundaries are sufficient for the use case in question. For clarity, we have emphasized these points in the main text in lines 419-422 (and please see regional classifications above).

*As a rough gauge of the temporal genetic stability of *P. vivax*, we used the mantel test to search for evidence of a correlation in parasite genetic distance (nucleotide differences) and temporal distance (difference in year of collection) in the *P. vivax* isolates from Thailand. We selected Thailand for the analysis as it provided the greatest density of vivax genomes over a minimum 10-year period (see list below), although the sample size in several years is still not ideal. For the 119 samples/years listed below, matrices of the genome-wide genetic distance (SNP difference) and the temporal distance (year difference) were created using the R ape package. The genetic and temporal distance matrices were correlated using the mantel test in the python-based scikit-bio package. No evidence of a significant correlation was observed (correlation coefficient = -0.019, $p=0.791$). These results do not support a substantial change in parasite genetic make-up over the period investigated in this low-endemic setting. However, we acknowledge that more sophisticated techniques are needed to evaluate this further in larger sample sets with more uniform sample size over the period of study.*

Year	N
2001	2
2003	5
2006	6
2007	10
2009	4
2010	9
2011	42
2012	28
2013	12
2014	1

Minor suggestions:

- 1. how many strains are there in polyclonal infections? How are the authors certain that there are only two strains in these samples?***

We apologise for any confusion regarding polyclonal infection. **We have not made any assumption that there are only 2 strains in polyclonal infections;** there may be any number of clones at various proportions to one another. We restricted our analyses to **biallelic SNPs (not samples)** to simplify data handling and processing, as is commonly done for *Plasmodium* studies (e.g., see <https://pubmed.ncbi.nlm.nih.gov/27348299/>), but this makes no assumption about the clonal composition of polyclonal infections.

- 2. sub labels need to be added on Figure 1 and referenced in the methods section where each part is described. Currently, it's very hard to know the correspondence between the flowchart and the section in the text.***

Thank you for pointing this out, we have revised the Figure as advised.

- 3. Axes need to be labeled in Supplementary Figure 1***

Thank you for pointing this out, we have revised the Figure as advised.

- 4. Line 84, "simulations" mentioned in the manuscript multiple times. However, it's unclear what kind of simulations are run. Random removal of genotypes in an empirical sample cannot be referred to as simulations.***

We do consider random removal of genotypes to be a simulation, but we are open to replacing this term with an alternative term suggested by the reviewer - please advise.

Reviewer #2 (Remarks to the Author):

The authors have used an innovative machine learning approach to select SNPs that can discriminate the country origin of P. vivax infections. These SNP barcodes are increasingly important in resolving sources and sinks of infections and hence useful for countries that are moving towards elimination. It is commendable that the work has also led to the development of an openly available tool for use by non experts in better understanding the sources of infections. The translation of the barcode and tool across the global P. vivax endemic population will depend on access to data generation facilities, in this case deep amplicon sequencing. It should be noted that deep sequencing technologies remain inaccessible to most national public health facilities and the malaria control programs. This does not minimise the validity and future usefulness of the tools developed here.

We thank the reviewer for their acknowledgment of the utility and importance of these tools to support the elimination of an important infectious disease. We have received similar enthusiasm from our partners in the vivax malaria community. Our collaborators at the Institute for Tropical Medicine have already established Illumina amplicon-based sequencing assays for the 33-SNP barcode (GEO33) (<https://pubmed.ncbi.nlm.nih.gov/36034708/>).

We also appreciate the reviewer's acknowledgement of our efforts to translate these tools into the real-world implementation setting. On this note, we can allay the reviewer's concerns about amplicon sequencing accessibility with reference to work being undertaken by our partners in the GenRe-Mekong project (<https://pubmed.ncbi.nlm.nih.gov/34372970/>) where **in-country amplicon-based sequencing is already being implemented in countries such as Vietnam for the surveillance of antimalarial resistance.**

We should also highlight that any markers selected within the described frameworks can be applied to other genotyping methods such as **minION sequencing**, which is designed for greater applicability in LMIC settings. We have added comments on the above points to the discussion in lines 432-435.

There are a number of issues that could be addressed to further improve the paper.

1. The study takes opportunity of freely available genome data and therefore has no control in the design of the original studies that generated the data. In this case, it should be noted that the sampling cannot be generalised to represent a country as the samples could be geographically biased even within country besides being designed to access a specific population sub-group such as clinical cases for therapeutic efficacy studies. Within country variation can be palpable in some settings such as Ethiopia where geneflow between regions are subject to restrictions due to the geography of the population.

2. The census numbers for some populations were small and as stated above can hardly be representative of the country of origin. Was the effect of this small numbers simulated

As mentioned in response to some of reviewer 1's queries, we acknowledge the constraints of the available genomic data, which is unfortunately beyond our control. However, **what we aim to showcase in this study is an important framework that will continually enhance as new data is added, and that**

can be adapted to other boundaries to serve user-defined use cases. We have clarified this in the text on lines 414-422 and have demonstrated the flexibility of the tools with the regional classification data shown above.

3. The GSK studies were selected for validation. These were from some of the countries included in the training set. Hence the samples are not that independent, though the source studies are independent. Will an iteration through several random subsets from different studies and source populations not increase the confidence and generalisation of the selected markers.

Our study used all data from the public domain that was available to us at the time of establishing the tools. But we agree with the reviewer on the importance of conducting continual evaluations to enhance the tools as new data is produced; **the frameworks have been intentionally designed to support this iterative process.**

4. The authors identified three new sets of markers upon the already existing 38 barcode from Broad Institute. The authors lean towards a 71 marker barcode when a new 33 SNPs are combined with the existing 38 and this seem to significantly improve the level of discrimination provide by either. However, the 50 and 55 barcodes perform comparably with these even with poor quality samples. Why was all combinations of set pairs not tested as for the 33 plus 38. I could not determine if each of the 4 sets where unique or there are overlapping SNPs in these different sets. Can there be a consensus set from combining all sets.

As the pre-existing 38-SNP Broad barcode (BR38) is already being used in programs such as the GenRe-Mekong program, we were aware of the need to provide an option for appending to (rather than replacing) this barcode. **We therefore considered the smallest new barcode selection as optimal for appending to the pre-existing Broad barcode owing to cost factors** (increasing numbers of SNPs bring increasing primer costs). For these reasons, our results/discussions did indeed have a focus on the 33-SNP barcode (GEO33). Nonetheless, we acknowledge the reviewer's interest in the performance of the 50-SNP and 55-SNP panels (GEO50 and GEO55) and have accordingly added further details on the performance of these panels in combination with BR38 (please see updated Figures 2-4 and updated Tables 1-2 and Supplementary Table 6). In brief, when there is no missing data (i.e., no genotyping fails), the country prediction accuracy of GEO50 and GEO55 are only slightly improved by adding BR38 (i.e., GEO50+BR38 and GEO55+BR38). However, when there are high levels of missing data, such as 30% of genotypes failing, GEO50 and GEO55 experience moderate improvement in performance from the addition of BR38, likely reflecting a degree of redundancy brought in by the extra SNPs.

We have also provided information on the overlap between the panels (lines 341-351, Supplementary Figure 5 and Supplementary Table 5). In brief, there is no overlap between BR38 and any of the GEO panels, but 22 SNPs overlap between two or more GEO panels. However, it should be noted that some of the overlapping SNPs may have redundancy (i.e., may distinguish the same countries), and the overlapping panel might not distinguish some countries well.

Reviewers' comments:

Reviewer #1 (Remarks to the Author):

I appreciated the detailed responses from the authors regarding my previous comments. It's now very clear and well written. I just have one general question, which is related to a technical question regarding the BALK classifier: the main text talked about the BALK can handle polyclonal infections. When I read the supplements, it seems that the classifier considers the scenario of diploid (two sets of chromosomes), does it mean that the classifier only considers MOI (or COI) = 2? It's not immediately clear to me whether the classifier handles more than two infections? And when it deals with polyclonal infections, does it predict their origin separately?

Another technical question: Naive Bayes is defined as setting a uniform prior, in this case the Naive Bayes is essentially equivalent to maximum likelihood, or am I missing something here?

Reviewer #2 (Remarks to the Author):

The additional supplementary material has added clarity to the methods and I am satisfied that this paper and its tools will be informative and useful to the field

Response to reviewers

*Note, line number given in our responses refer to the revised manuscript (i.e., with track changes)

Reviewer #1 (Remarks to the Author):

- 1. I appreciated the detailed responses from the authors regarding my previous comments. It's now very clear and well written. I just have one general question, which is related to a technical question regarding the BALK classifier: the main text talked about the BALK can handle polyclonal infections. When I read the supplements, it seems that the classifier considers the scenario of diploid (two sets of chromosomes), does it mean that the classifier only considers MOI (or COI) = 2? It's not immediately clear to me whether the classifier handles more than two infections? And when it deals with polyclonal infections, does it predict their origin separately?***

We thank the reviewer for their constructive feedback, which has greatly helped to improve the manuscript. Regarding the query about polyclonal infections, this is a good question that we have attempted to address more clearly in revisions of the main text and supplementary information as outlined below. We hope this clarifies any confusion for the reviewer and potentially other readers.

Min text, lines 185-187: *The restriction to bi-allelic SNPs is a standard approach undertaken in malaria population genomics to simplify downstream computations and does not impose constraints on the analysis of polyclonal infections, which are still detectable through the composite of allelic variants across the respective SNPs (see ¹⁷⁻¹⁹).*

Main text, lines 471-484: *The likelihood-based classifier framework has been designed to allow geographic predictions to be assigned to polyclonal infections carrying two or more clones, as are common in high endemicity regions; these infections are commonly omitted from population genetic analyses. However, it should be acknowledged that the classifier does not attempt to phase individual clones, rather the infection is analyzed as a composite, yielding a single prediction of most likely origin. Nonetheless, it is important to note that, by design, the GEO panels selected by the framework should exhibit low within-country diversity (with diversity rather being between countries). Polyclonal infections deriving from a single country should therefore exhibit a low frequency of heterozygote positions at the selected GEO barcodes. In cases where a combination of clones deriving from different countries are present within a single infection, yielding many heterozygote positions, the classifier will be constrained in its ability to detect country of origin and a low confidence in the prediction will accordingly be assigned. Future developments that combine GEO markers with high-resolution finger-printing markers such as microhaplotypes may enable polyclonal infections to be phased and subsequently analyzed for geographic origin.*

Supplementary, lines 272-279: *One of the requirements for the classifier in this analysis is the ability to analyze heterozygous alleles. Heterozygous alleles can arise in samples having polyclonal infection of 2 or more strains. If each sample can be represented as diploid with 2 sets of chromosomes regardless of its polyclonality, then obtaining the allele from a SNP position out of those 2 sets of chromosomes can be thought as 2 consecutive independent experiments, and hence a binomial distribution with $N=2$ can be*

used for the outcome probability of this data. The SNP data then can be stated as the number of occurrences of alternate allele in each respective position with homozygous reference allele stated as 0 (0 alternate allele, 2 reference alleles), heterozygous as 1 (1 alternate allele, 1 reference allele), and homozygous alternate allele as 2 (2 alternate alleles, 0 reference allele).

2. Another technical question: Naive Bayes is defined as setting a uniform prior, in this case the Naive Bayes is essentially equivalent to maximum likelihood, or am I missing something here?

Indeed, when a Naive Bayes method has its prior probability set to a uniform distribution, its posterior probability will only depend on its likelihood function (line 245-246 of Supplementary Material), hence its maximum posterior probability is equivalent to its maximum likelihood. Future developments can introduce non-uniform prior probability in scenarios such as when a sample has robust travel history information (definition of “robust” yet to be defined), which can be used to increase the odds of the travel history-based country prediction by adjusting the prior probability of the respective countries accordingly. We have added clarification to the Supplementary Material as outlined below.

Supplementary, lines 243-245: Hence, for the purpose of classifying by country of origin, the prior probability will be set to uniform distribution (uniform values for all countries) so it can be dropped from the equation and that the classification rule only depends on the likelihood part of the formula, which effectively makes the classification rule equivalent to maximum likelihood.